# Mitochondrial DNA Changes in Blood and Urine Display a Specific Signature in Relation to Inflammation in Normoalbuminuric Diabetic Kidney Disease in Type 2 Diabetes Mellitus Patients

**DOI:** 10.3390/ijms24129803

**Published:** 2023-06-06

**Authors:** Ligia Petrica, Adrian Vlad, Florica Gadalean, Danina Mirela Muntean, Daliborca Vlad, Victor Dumitrascu, Flaviu Bob, Oana Milas, Anca Suteanu-Simulescu, Mihaela Glavan, Dragos Catalin Jianu, Sorin Ursoniu, Lavinia Balint, Maria Mogos-Stefan, Silvia Ienciu, Octavian Marius Cretu, Roxana Popescu

**Affiliations:** 1Department of Internal Medicine II, Division of Nephrology, “Victor Babes” University of Medicine and Pharmacy, No. 2, Eftimie Murgu Sq., 300041 Timisoara, Romania or petrica.ligia@umft.ro (L.P.); bob.flaviu@umft.ro (F.B.); oana.milas@yahoo.com (O.M.); anca.simulescu@yahoo.com (A.S.-S.); mihaelapatruica@gmail.com (M.G.); lavinia.balint@umft.ro (L.B.); maria.stefan2014@yahoo.com (M.M.-S.); ienciu.silviaoana@yahoo.com (S.I.); 2Centre for Molecular Research in Nephrology and Vascular Disease, Faculty of Medicine, “Victor Babes” University of Medicine and Pharmacy, No. 2, Eftimie Murgu Sq., 300041 Timisoara, Romania; vlad.adrian@umft.ro (A.V.); daninamuntean@umft.ro (D.M.M.); vlad.daliborca@umft.ro (D.V.); dumitrascu.victor@umft.ro (V.D.); jianu.dragos@umft.ro (D.C.J.); sursoniu@umft.ro (S.U.); popescu.roxana@umft.ro (R.P.); 3Centre for Cognitive Research in Neuropsychiatric Pathology (Neuropsy-Cog), Faculty of Medicine, “Victor Babes” University of Medicine and Pharmacy, No. 2, Eftimie Murgu Sq., 300041 Timisoara, Romania; 4Center for Translational Research and Systems Medicine, Faculty of Medicine, “Victor Babes” University of Medicine and Pharmacy, No. 2, Eftimie Murgu Sq., 300041 Timisoara, Romania; 5County Emergency Hospital Timisoara, 300723 Timisoara, Romania; 6Department of Internal Medicine II, Division of Diabetes and Metabolic Diseases, “Victor Babes” University of Medicine and Pharmacy, No. 2, Eftimie Murgu Sq., 300041 Timisoara, Romania; 7Department of Functional Sciences III, Division of Pathophysiology, “Victor Babes” University of Medicine and Pharmacy, No. 2, Eftimie Murgu Sq., 300041 Timisoara, Romania; 8Department of Biochemistry and Pharmacology IV, Division of Pharmacology, “Victor Babes” University of Medicine and Pharmacy, No. 2, Eftimie Murgu Sq., 300041 Timisoara, Romania; 9Department of Neurosciences VIII, Division of Neurology I, “Victor Babes” University of Medicine and Pharmacy, No. 2, Eftimie Murgu Sq., 300041 Timisoara, Romania; 10Department of Functional Sciences III, Division of Public Health and Health and History of Medicine, “Victor Babes” University of Medicine and Pharmacy, No. 2, Eftimie Murgu Sq., 300041 Timisoara, Romania; 11Department of Surgery I, Division of Surgical Semiology I, “Victor Babes” University of Medicine and Pharmacy, No. 2, Eftimie Murgu Sq., 300041 Timisoara, Romania; tavicretu@yahoo.com; 12Emergency Clinical Municipal Hospital Timisoara, 300041 Timisoara, Romania; 13Department of Microscopic Morphology II, Division of Cell and Molecular Biology II, “Victor Babes” University of Medicine and Pharmacy, No. 2, Eftimie Murgu Sq., 300041 Timisoara, Romania

**Keywords:** mitochondrial DNA, inflammation, podocyte, proximal tubule, normoalbuminuria, diabetic kidney disease, type 2 diabetes mellitus

## Abstract

Mitochondrial dysfunction is an important mechanism contributing to the development and progression of diabetic kidney disease (DKD). Mitochondrial DNA (mtDNA) levels in blood and urine were evaluated in relation to podocyte injury and proximal tubule (PT) dysfunction, as well as to a specific inflammatory response in normoalbuminuric DKD. A total of 150 type 2 diabetes mellitus (DM) patients (52 normoalbuminuric, 48 microalbuminuric, and 50 macroalbuminuric ones, respectively) and 30 healthy controls were assessed concerning the urinary albumin/creatinine ratio (UACR), biomarkers of podocyte damage (synaptopodin and podocalyxin), PT dysfunction (kidney injury molecule-1 (KIM-1) and *N*-acetyl-β-(D)-glucosaminidase (NAG)), and inflammation (serum and urinary interleukins (IL-17A, IL-18, and IL-10)). MtDNA-CN and nuclear DNA (nDNA) were quantified in peripheral blood and urine via qRT-PCR. MtDNA-CN was defined as the ratio of the number of mtDNA/nDNA copies via analysis of the CYTB/B2M and ND2/B2M ratio. Multivariable regression analysis provided models in which serum mtDNA directly correlated with IL-10 and indirectly correlated with UACR, IL-17A, and KIM-1 (R^2^ = 0.626; *p* < 0.0001). Urinary mtDNA directly correlated with UACR, podocalyxin, IL-18, and NAG, and negatively correlated with eGFR and IL-10 (R^2^ = 0.631; *p* < 0.0001). Mitochondrial DNA changes in serum and urine display a specific signature in relation to inflammation both at the podocyte and tubular levels in normoalbuminuric type 2 DM patients.

## 1. Introduction

Diabetic kidney disease (DKD) represents the most important cause of end-stage renal disease worldwide, accounting for over 40% of diabetic patients with both type 1 and type 2 diabetes mellitus (DM) undergoing renal replacement therapies [1].

The mechanisms of albuminuria processing in the course of DM involve alterations to all structures of the glomerular filter, especially podocytes, but also to other glomerular cells. In the last decade, the tubular theory concerning the mechanisms of albuminuria in DKD emphasizes the central stage role of proximal tubule (PT) dysfunction in the occurrence of albuminuria within the confines of diabetic tubulopathy [2,3]. The tubular theory has been used to challenge the glomerular view of albuminuria due to robust evidence in favor of there being a significant number of patients with DM who will develop progressive decline in renal function in the absence of albuminuria. The newest category, which comprises approximately 60% of the patients with type 2 DM [4] and 60% of the patients with type 1 DM [5], is referred to as normoalbuminuric DKD. 

Normoalbuminuric DKD, which has the most significant prevalence to date [6], may be defined not only in relation to the urinary albumin/creatinine ratio < 30 mg/g and e GFR, but also to biomarkers of PT dysfunction. It has been postulated that PT dysfunction is a more accurate predictor of DKD progression than glomerular involvement is [3,7]. This pivotal role of the PT is supported by studies that show that the number of PT dysfunction biomarkers increases before the occurrence of albuminuria, an observation that is in favor of the fact that a proximal tubular injury precedes the appearance of glomerular lesions [8,9,10,11].

Multiple mechanisms are involved in the development and progression of DKD. Emerging data show that inflammatory pathways have a well-defined role, which involves interacting with various molecular mechanisms at the glomerular and tubular levels [12]. The number of proinflammatory cytokines, as effector molecules of inflammatory processes, is increased in DKD, and they are associated with both glomerular damage and PT dysfunction biomarkers, even in the normoalbuminuric stage of DKD [13,14]. The increase in the number of proinflammatory cytokines, such as IL-17A [15] and IL-18 [16,17], and the decrease in the number of anti-inflammatory cytokines, such as IL-10, even in the absence of micro- and macroalbuminuria, respectively, are consistent with this observation [18].

In addition to the glomerular theory and the tubulocentric theory concerning the mechanisms of albuminuria during the course of DKD, the mitochondria-centric view of DKD has gained attention due to the fact that mitochondria, as complex organelles that are present in all cells, except erythrocytes, are abundantly found in the kidneys, both within PT cells and the glomerular cells, such as the podocytes, mesangial cells, and endothelial cells [19,20]. 

Mitochondrial dysfunction is an important factor contributing to the development and progression of DKD [21]. This implies multiple alterations of complex mitochondrial functions, including the generation of altered mitochondrial deoxyribonucleic acid (mtDNA) under excessive oxidative stress conditions, is of major importance [19,22].

Furthermore, mitochondrial bioenergetics abnormalities may trigger inflammatory mechanisms. which may cause chronic inflammation, renal tissue injury [20], and damage to mtDNA [23]. Mitochondrial DNA changes and the complex inflammatory processes involved in the pathogenesis of DKD precede histological and biochemical changes in DKD, and they may be detected early on during the course of DKD, before the occurrence of albuminuria [24,25].

The aim of the study was to evaluate the potential relationship between mtDNA modifications in blood and urine and podocyte injuries and PT dysfunction in early-stage DKD in type 2 DM patients. In addition, we attempted to verify the hypothesis according to which mtDNA levels in serum and urine may be related to a specific inflammatory response in normoalbuminuric DKD in type 2 DM patients.

## 2. Results

### 2.1. Demographic, Clinical, and Biological Data of Patients with Type 2 DM and Healthy Control Subjects

Demographic, clinical, and biological data about patients with type 2 DM and healthy controls are presented in Table 1 as medians and IQR. In Table 1, the *p* values were corrected for multiple comparisons. These data show significant differences between the groups studied. There were decreased levels of serum mtDNA and of anti-inflammatory serum and urinary IL-10, as well increased levels of urinary mtDNA, serum and urinary proinflammatory cytokines (IL-17A and IL-18), podocyte damage biomarkers (urinary synaptopodin and podocalyxin), and of biomarkers of PT dysfunction (urinary KIM-1 and NAG) in type 2 DM patients (from normal to mildly increased albuminuria and from moderately increased albuminuria to severely increased albuminuria) vs. healthy control subjects. 

### 2.2. MtDNA Changes Correlate with Podocyte Injuries and PT Dysfunction in Early-Stage DKD in Type 2 DM Patients

In our study, the serum mtDNA level was decreased, while the urinary mtDNA level was increased among all patient groups compared to that of the healthy controls. In univariable regression analysis, serum mtDNA directly correlated with eGFR and negatively correlated with UACR, urinary synaptopodin, urinary podocalyxin, urinary KIM-1, and urinary NAG.

Urinary mtDNA correlated directly with UACR, synaptopodin, podocalyxin, KIM-1, and NAG and showed an inverse correlation with eGFR (Table 2).

These results are indicative of the involvement of mtDNA changes in functional modifications within podocytes and proximal tubules in the early stages of DKD, a fact which derives from the variations in biomarkers of podocyte damage and PT dysfunction.

### 2.3. MtDNA Levels in Serum and Urine Are Associated with a Specific Inflammatory Response in Normoalbuminuric DKD in Type 2 DM Patients

Serum mtDNA negatively correlated with serum and urinary IL-17A and with serum and urinary IL-18 and directly correlated with serum and urinary IL-10.

Urinary mtDNA correlated directly with IL-17A and IL-18 and had an inverse correlation with IL-10 (Table 3).

Proinflammatory cytokines IL-17A and IL-18 were highly expressed in type 2 DM patients, and their levels in blood and urine followed an ascending trend from normal/mildly increased albuminuria to severely increased albuminuria, while IL-10, an anti-inflammatory cytokine, displayed a descending trend. This inflammatory profile was associated with mtDNA changes in serum and urine, namely, decreased serum mtDNA and increased urinary mtDNA levels, which initiated either a proinflammatory response, as shown by the increased levels of IL-17A and IL-18, or an anti-inflammatory response, namely, the decreased expression of IL-10.

Multivariable regression analysis yielded models in which serum mtDNA directly correlated with IL-10 and negatively correlated with UACR, IL-17A, and KIM-1 (R^2^ = 0.626; *p* < 0.0001), while urinary mtDNA had a direct correlation with UACR, podocalyxin, NAG, and IL-18, and an indirect correlation with IL-10 and eGFR (R^2^ = 0.631; *p* < 0.0001). The values of R^2^ show that there are strong correlations between mtDNA and the variables studied (Table 4). 

## 3. Discussion

In the current research, we studied a potential relationship between mtDNA modifications and podocyte injuries and proximal tubule dysfunction in early-stage DKD in type 2 DM patients. In addition, we attempted to verify the hypothesis according to which mtDNA changes in blood and urine may be related to a specific inflammatory response in normoalbuminuric DKD in type 2 DM patients. Our observations point to the fact that the deregulated mtDNA pattern parallels inflammation both at the glomerular and tubular levels even in normoalbuminuric DKD in type 2 DM patients.

### 3.1. MtDNA Changes Impact Both Podocytes and Proximal Tubules in Normoalbuminuric DKD in Type 2 DM Patients

In contrast to nuclear DNA, human mtDNA is a circular molecule made up of 16,569 base pairs and composed of two strands (heavy and light strands), which encode essential protein subunits of the OXPHOS system within the mitochondria. MtDNA is highly susceptible to oxidative stress due to the lack of histone protection and defense mechanisms in the coding regions [26]. 

Damaged mitochondria release their content into the extracellular space and into systemic circulation afterwards [27]. Thus, mtDNA fragments deriving from systemic circulation are filtered through the glomeruli and actively secreted into urine. Extracellular mtDNA may be detected and quantified in both serum and urine, serving as a biomarker of mitochondrial dysfunction. Urinary mtDNA originating from injured renal tissue has been a relevant biomarker indicative of various renal diseases. Thus, the level urinary mtDNA is increased within the confines of an acute kidney injury, either in experimental models in mice [28] or in human studies [29]. Additionally, urinary mtDNA emerged as a reliable biomarker of renal function and a renal tissue injury in a study conducted on patients with biopsy-proved hypertensive nephrosclerosis and IgA nephropathy [30], as well as in a study performed on patients with non-diabetic kidney disease [31].

Several studies showed that patients with type 2 DM and DKD had lower levels of serum mtDNA than the diabetic controls did [26]. The latter group had increased levels of serum mtDNA, most likely as an adaptative response to a metabolic-induced long-term mitochondrial injury during the course of type 2 DM [26]. Additionally, it was underlined that DKD patients had decreased mtDNA copy numbers in serum, which is associated with increased mtDNA damage [32].

In our study, we found decreased serum mtDNA levels among all patient groups compared to that of the healthy controls, which is in keeping with previous studies performed on type 2 DM patients [22,32]. Furthermore, our study provided results that show increased levels of urinary supernatant mtDNA, which displayed an ascending trend from normo- to macroalbuminuria type 2 DM patients. Similar data were reported by the authors of other studies, which found increased levels of urinary supernatant mtDNA in type 2 DM patients [33]. The authors queried the origin of urinary mtDNA, namely whether an increased mtDNA level in urine is related to the continuous destruction of mitochondrial content within the kidney rather than to the increased clearance of circulating mtDNA [33]. Studies focused on alterations of serum mtDNA or urinary mtDNA did not include correlations with renal histology, except for a study by Wei et al. [33]. In this study, the authors showed that urinary supernatant mtDNA had a significant inverse correlation with intra-renal mtDNA and correlated with the severity of tubulointerstitial fibrosis, but not with glomerulosclerosis according to a biopsy, even in early stages of diabetic nephropathy [33].

In our study, the levels of urinary supernatant mtDNA showed a direct correlation with albuminuria across all patient groups and healthy controls and an inverse correlation with eGFR. Of note, these correlations were more significant in the normoalbuminuric group compared to that of the healthy controls. In our normoalbuminuric group, the direct correlation of urinary mtDNA with biomarkers of PT dysfunction may also explain the direct correlation of urinary mtDNA with albuminuria due to the fact that albuminuria processing is highly dependent on the PT during the course of type 2 DM. The inverse correlation of urinary mtDNA with eGFR could be attributed to a progressive decline in renal function, in parallel with the incapacity of the PT to retrieve albumin in the early stages of DKD. Urinary mtDNA may originate from both circulating mtDNA, which is subsequently filtered by the kidney, and intra-renal mtDNA, which results from a renal tissue injury. In our study, due to the lack of renal tissue analysis, we could neither assess the correlation of the urinary mtDNA level with intra-renal mtDNA content, nor the correlation of eGFR with intra-renal mtDNA.

In the study by Wei et al., the authors remarked that higher levels of urinary supernatant mtDNA correlated with a lower baseline eGFR [33]. Additionally, the authors underline the fact that even the lowest concentration of urinary supernatant mtDNA, namely tertile I, among patients that were considered to have early stages of diabetic nephropathy and mild mitochondrial dysfunction, had a modest correlation with proteinuria and a marginal correlation with eGFR [33].

Due to the scarce amount of data in the literature with regard to the impact of mtDNA abnormalities on renal tissue, we assessed the biomarkers of podocyte damage and PT dysfunction in order to evaluate the potential relationship between serum and urinary mtDNA changes and glomerular and tubular lesions. We found a significant inverse correlation between podocyte biomarkers synaptopodin and podocalyxin and between PT dysfunction biomarkers, urinary KIM-1 and NAG, and serum mtDNA, as well as a direct correlation of these biomarkers with urinary mtDNA. These data suggests that increased urinary mtDNA levels may be attributed to increased filtration of circulating DNA, according to the hypothesis that increased levels of urinary mtDNA may derive from filtered circulating mtDNA due to lesions on the glomerular filter associated with PT dysfunction. However, in our study, due to a lack of renal tissue analysis, impairment to intra-renal mtDNA cannot be excluded.

### 3.2. Alterations in mtDNA Profile and the Associated Inflammatory Processes Occur Early on during the Course of Type 2 DM

Another objective within the confines of the above-mentioned changes in serum and urinary mtDNA was to evaluate the relationship between the changes in the mtDNA profile and inflammatory processes, which participate in the development of DKD. Inflammatory mediators, such as proinflammatory cytokines, are involved in the initiation and progression of DKD.

Interleukin-17A (IL-17A) is a member of IL-17 family produced by activated T helper (Th) lymphocytes Th17, macrophages, neutrophils, natural killers, dendritic, and mast cells. IL-17A performs pleiotropic functions via inducing the expression of other proinflammatory cytokines, which cause tissue inflammation [34].

In our study, serum and urinary IL-17A levels directly correlated with albuminuria and biomarkers of podocyte damage and PT dysfunction and negatively correlated with eGFR. Our findings are supported by a study conducted by Ma et al., in which IL-17A increased the expression of proinflammatory cytokines IL-6 and TNF-α in podocytes in vitro and in cultured tubular epithelial cells [35]. Moreover, the transcript level of IL-17A was increased in the blood of type 2 DM patients during all five stages of chronic kidney disease. Of interest, these changes occurred as early as stages 1–2, before a significant decline in renal function [36]. In an experimental study on mice, IL-17A blockade elicited a significant decrease in the level of albuminuria and an improvement in renal lesions [34]. Additionally, a treatment with IL-17A-neutralizing antibody diminished the renal gene expression of KIM-1 and neutrophile gelatinase-associated lipocalin (NGAL) and restored both the podocyte number and podocin gene expression level [34]. The results of our study showed that serum and urinary IL-17A levels negatively correlated with serum mtDNA and directly correlated with urinary mtDNA, an observation that might point to the potential relationship between this proinflammatory cytokine and mtDNA changes. The literature is lacking data with regard to this issue, but given the correlations of IL-17A with albuminuria, eGFR, biomarkers of podocyte injury, and PT dysfunction, on the one hand, and with the deregulated profile of serum and urinary mtDNA on the other, we may presume that mtDNA might derive from injured podocytes and proximal tubular cells and could intervene in modulating IL-17A activity.

Interleukin-18 (IL-18) is a proinflammatory cytokine that is constitutively expressed by tubular epithelial cells, which is strongly implicated in the initiation and progression of DKD [37] via various mechanisms, such as the activation of other inflammatory mediators, apoptosis, and oxidative stress [12,16]. Additionally, increased plasma and urinary levels of IL-18 are associated with early stages of diabetic nephropathy, as shown by its correlations with high-to-normal levels of albuminuria [38,39]. In our study, serum and urinary IL-18 levels were increased in all patient groups studied and positively correlated with albuminuria and the biomarkers of podocyte damage and PT dysfunction and negatively correlated with eGFR, as shown by us in previous studies [13,14]. These results indicate that increased levels of serum and urinary IL-18 are present even in normoalbuminuric patients and induce the impairment of both podocytes and PTs in the early stage of DKD. Similar data were shown by the authors of other studies, who found that serum IL-18 may be a predictor of future renal dysfunction in normoalbuminuric type 2 DM patients [40]. The impairment of mtDNA induces the activation of various inflammatory pathways, leading to chronic inflammatory processes within the kidney in diabetic nephropathy [22]. Our study showed that serum and urinary IL-18 levels correlated with serum and urinary mtDNA levels, which in turn, correlated with albuminuria, eGFR, podocyte injury, and PT dysfunction biomarkers. This association of events might allow for the speculation that mtDNA variations in serum and urine may trigger the increased expression of IL-18 and subsequent podocytes and PT involvement. Defective OXPHOS and increased ROS production induce mutations in mtDNA, which further promote the production of ROS via the impairment of OXPHOS [19]. ROS cause downstream inflammatory cascades and the release of proinflammatory cytokines, including pro-interleukin-18 [23]. 

Interleukin-10 (IL-10) is an anti-inflammatory cytokine that can reduce the inflammatory infiltration of renal tissue, thus alleviating the proliferation of mesangial cells and interstitial fibrosis [41]. Furthermore, IL-10 polymorphisms protect type 2 DM patients against the risk of DKD [42] and display multiple renoprotective effects after diabetic myocardial infarction [43].

In our study, serum and urinary levels of IL-10 negatively correlate with albuminuria and the biomarkers of podocyte injury and PT dysfunction and directly correlate with eGFR, which shows the protective role of IL-10 in the early stages of DKD. It has been demonstrated that urinary IL-10 allows the identification of DKD, even in the absence of micro- and macroalbuminuria [18]. Additionally, our study revealed the fact that serum and urinary IL-10 levels correlated with mtDNA changes, which may trigger variations in the expression of IL-10 both at the glomerular and tubular levels, as assumed from the correlations of podocyte and PT biomarkers with IL-10 and mtDNA modifications. Circulating cell-free mtDNA is known as a mitochondrial damage-associated molecular pattern, which could stimulate a cellular anti-inflammatory response [44] and subsequently cause renal tissue lesions [45]. The increased mtDNA-induced expression of IL-10 as a defense mechanism against an injury could explain the association of serum and urinary mtDNA with variations in the expression of podocyte damage and PT dysfunction biomarkers among all patient groups, even in normoalbuminuric type 2 DM patients. The lack of IL-10 enhances the activation of fibrotic cytokines and produces tubular lesions and interstitial fibrosis [46].

The results of our study point to a perspective view with regard to a complex interrelationships between mtDNA changes and concomitant pro- and anti-inflammatory mechanisms.

Our study has several limitations. First, this is a cross-sectional study, in which we only studied associations between the phenomena observed and not the relation of causality between mtDNA changes and inflammatory mechanisms in the pathogenesis of DKD. Second, the study did not include renal biopsies and renal tissue analysis both at the glomerular and tubular levels with regard to the parameters studied. MtDNA oxidative stress damage in glomeruli and oxidized lesions within glomerular endothelial cells were not studied. These data, in addition to DNA adducts evaluation, would have allowed us to assess the mtDNA abnormalities involved in DKD pathogenesis and progression [47]. Third, in addition to mtDNA quantitative assessment, the reactive oxygen species profile, which could result in mtDNA impairment, was not studied. 

The strengths of our study reside in the demonstration of mtDNA variations in serum and urine associated with a specific inflammatory response in early-stage DKD in type 2 DM patients. Additionally, we attempted to provide an integrated view of mitochondrial dysfunction in which the deregulated pattern of mtDNA in serum and urine parallels inflammation both at the glomerular and tubular levels in normoalbuminuric type 2 DM patients.

## 4. Materials and Methods

### 4.1. Patients’ Selection

A total of consecutive 220 patients with type 2 DM attending the Outpatient Department of Nephrology and the Outpatient Department of Diabetes and Metabolic Diseases, County Emergency Hospital Timisoara, Romania (from January 2021 to December 2022) aged between 50 and 78 years were screened for the study according to person visits and chart reviews. The inclusion criteria were: having a DM duration higher than 5 years and undergoing therapy with oral antidiabetic drugs (metformin and gliclazide) or insulin and angiotensin-converting enzyme inhibitors or angiotensin receptor blockers and statins. The exclusion criterion was having a decompensated DM (HbA_1c_ > 10%). A total of 150 patients (52 patients with normoalbuminuria (group 1, urinary albumin: creatinine ratio-UACR < 30 mg/g), 48 patients with moderately increased albuminuria (group 2, UACR 30–300 mg/g), and 50 patients with severely increased albuminuria (group 3, UACR > 300 mg/g)) and 30 age- and gender-matched healthy control subjects who attended a general practitioner’s office for routine check-up, without a known history of renal diseases and in whom DM or pre-diabetes were excluded by having a value of HbA_1c_ ≤ 5.6% (group 4) were consecutively enrolled in this case series study. The remaining 70 patients out of the 220 patients screened were excluded due to their HbA_1c_ levels.

### 4.2. Laboratory Assessments

The serum and urinary specimens of patients and controls were frozen at −80 °C and thawed before assaying. Urinary biomarkers were evaluated in the same first morning urine sample and were reported as the per urinary creatinine ratio. The ELISA technique was utilized for the assessment of podocyte damage biomarkers: synaptopodin (Catalogue Nr. abx055120; Abbexa; sensitivity—0.10 ng/mL; detection range—0.156–10 ng/mL; coefficient of variance (CV) < 10%), podocalyxin (Catalogue Nr. E-EL-H2360; Elabscience; sensitivity—0.1ng/mL; detection range—0.16–10 ng/mL; coefficient of variance (CV) < 10%); PT dysfunction biomarkers: KIM-1 (Catalogue Nr. E-EL-H6029; Elabscience; sensitivity—4.69 pg/mL; detection range—7.81–500 pg/mL; CV < 10%), NAG (Catalogue Nr. E-EL-H0898; Elabscience; sensitivity—0.94 ng/mL; detection range—1.56–100 ng/mL; CV < 10%); interleukins: IL-17A (Catalogue Nr. E-EL-H0105; Elabscience; sensitivity—18.75 pg/mL; detection range—31.25–2000 pg/mL; CV < 10%), IL-18 (Catalogue Nr. E-EL-H0253; Elabscience; sensitivity—9.38 pg/mL; detection range—15.63–1000 pg/mL; CV < 10%), and IL-10 (Catalogue Nr. E-EL-H6154; Elabscience; sensitivity—0.94 pg/mL; detection range—1.56–100 pg/mL; CV < 10%). All serum and urinary samples were run in triplicate and assessed according to the instructions of the manufacturer’s brochure. Urinary tract infections were ruled out if al patients had negative urine cultures. DKD was defined according to the consensus report by the American Diabetes Association (ADA) and Kidney Disease: Improving Global Outcomes (KDIGO) [7]; eGFR was calculated using the CKD-EPI equation, which includes serum creatinine and cystatin C, according to the KDIGO Guideline for the Evaluation and Management of Chronic Kidney Disease [48].

### 4.3. MtDNA Assessment

MtDNA-CN and nuclear DNA (nDNA) were quantified in peripheral blood and urine via qRT-PCR (CFX Connect-Biorad Laboratories, Carlsbad, CA, USA). TaqMan assays were utilized for the assessment of cytochrome b (CYTB) gene, subunit 2 of NADH dehydrogenase (ND2), and of beta 2 microglobulin nuclear gene (B2M). Primers for CYTB and ND2 genes were utilized as target sequences for the assessment of mtDNA. For nDNA analysis, B2M was utilized as an internal reference gene. MtDNA-CN was defined as the ratio of the number of mtDNA/nDNA copies via the analysis of the CYTB/B2M and ND2/B2M ratio. Genomic DNA was obtained from biological samples with the PureLink™ Genomic DNA Kit (Life technologies, Carlsbad, CA, USA) following the manufacturer’s recommendations. The concentration of extracted DNA was measured via fluorimetric quantification (Qubit, Invitrogen, Thermo Fisher Scientific, Waltham, MA, USA). For real-time quantitative tests, DNA samples were diluted to 10 ng/µL. A real-time quantitative polymerase chain reaction was induced using TaqMan Universal PCR master mix and TaqMan primers (Applied Biosystems, Thermo Fisher Scientific, Waltham, MA, USA). Samples were run in triplicate in MicroAmp^®^Optical 96-well reaction plates, and each well contained 9 µL of diluted DNA, 1 µL primers, and 10 µL of master mix. The total reaction volume was 20 μL. The thermal cycle profile was 2 min at 500C for UNG incubation, 10 min at 950C for polymerase activation, and 40 cycles of 15 s at 950C (denaturation), and 1 min at 600C (annealing and extension). For each run, we performed melting curve analysis to check for nonspecific products.

Relative mtDNA quantification was carried out as previously described [49]. The results were analyzed using the comparative Ct method. ΔCt (values of Δ cycle thresholds) in the sample were calculated by subtracting the values for the reference gene from the sample Ct and normalizing to nuclear DNA. 2-ΔCt was obtained, and the results are expressed as relative quantification. Obtained values were normalized to nuclear DNA and are reported as number of copies per nuclear DNA (mtDNA/nDNA). Urinary values were normalized to urine creatinine.

### 4.4. Statistical Analysis

Clinical and biological data are presented as medians and interquartile range (IQR) for variables with skewed distribution. Depending on the distribution of the values, differences between subgroups were analyzed using the Mann–Whitney U test for the comparison of 2 groups and the Kruskal–Wallis test for the comparison of 4 groups. Statistical methods were utilized as required in a case series study, which is an exploratory study. Case series have a descriptive study design. In our case series, analyses were performed in order to investigate the association between mtDNA changes in serum and urine and a particular inflammatory profile. Regression analyses were conducted in order to assess the significance of the relationship between serum and urinary mtDNA and serum and urinary interleukins, as well as with other continuous variables, such as UACR, eGFR, synaptopodin, podocalyxin, KIM-1, and NAG. Univariable regression analyses were carried out to evaluate the significance of the relationship between continuous variables for all 4 groups together (pooled data of normo- and micro- and macroalbuminuric patients and healthy controls). Only significant variables yielded via univariable regression analysis were introduced in the models for multivariable regression analysis.

We calculated the variance inflation factors (VIF) for each of the independent variables to assess the presence of multi-collinearity. Finally, we used Cameron and Trivedi’s test for heteroskedasticity. We measured the extent of multicollinearity among the independent variables for both models. Since all of the VIF values raged between 2 and 3, there is no problem of multicollinearity among the included variables in the models, and therefore, we maintained all our variables for the purpose of estimation.

One of the underlying assumptions of a linear regression model is that of homoskedasticity; so, the presence of heteroskedasticity in any form violates this assumption. The Cameron–Trivedi heteroskedasticity test was statistically non-significant for both models (*p* > 0.05); so, we accept the null hypothesis and conclude that homoskedasticity is present.

Statistical significance was set at *p* < 0.05, and the analyses were conducted with Stata 17.1 (Statacorp, College Station, TX, USA).

## 5. Conclusions

In conclusion, alterations in the mtDNA profile and the associated inflammatory processes occur early on during the course of type 2 DM. This association suggests the clinical applicability of the simultaneous identification of an interleukin profile and of mtDNA content in serum and urine using a non-invasive approach in order to increase the accuracy of the diagnosis of normoalbuminuric DKD patients. Mitochondrial DNA levels in serum and urine display a specific signature in relation to inflammation at the podocyte and tubular levels in normoalbuminuric type 2 DM patients. Further, longitudinal studies on larger cohorts are needed in order to prove causality between serum and urinary mtDNA changes and inflammation in normoalbuminuric type 2 DM patients.

## Figures and Tables

**Table 1 ijms-24-09803-t001:** Demographic, clinical, and biological data of patients with type 2 DM and healthy controls.

Parameter	Healthy Controls (N = 30)	Normal to Mildly Increased Albuminuria (A1) (N = 52)	Moderately Increased Albuminuria (A2) (N = 48)	Severely Increased Albuminuria (A3) (N = 50)
**Age (years)**	67.47 (64; 69)	68.33 (65; 72)	69.23 (65; 74)	69.8 (67; 73)
**BMI**	25.17 (23; 27) ^#,^*	29.26 (26.5; 31.5) ^⌂^	31.47 (28; 34)	30.8 (27; 32)
**SBP (mmHg)**	117.17 (110; 120) ^#,^*	138.37 (120; 150)	141.42 (130; 152.5)	147.8 (140; 165)
**DBP (mmHg)**	69 (65; 70) ^#,^*	79.15 (70; 90)	80.2 (70; 90)	81.5 (70; 90)
**DM duration (years)**	-	15.22 (10; 16.5)	17.73 (12; 23) ♦	21.04 (16; 26)
**Hb (g/dL)**	13.68 (13.1; 14) *	13.5 (12.75; 14.5) ^♣^	12.4 (11.46; 13.1)	12.39 (11.4; 13.4)
**Serum creatinine (mg/dL)**	0.8 (0.74; 0.85) ^#,^*	0.93 (0.86; 1) ^▲^	0.98 (0.9; 1.02) ^■^	1.33 (1.15; 1.5)
**Cystatin C (mg/L)**	0.9 (0.8; 1) *	0.95 (0.82; 1.1)	1 (0.88; 1.13) ^■^	1.25 (1.12; 1.38)
**eGFR (ml/min/1.73 m^2^)**	84.12 (80.64; 87.85) ^#,^*	76.2 (71.52; 81.21) ^♣^	70.53 (66.88; 73.94) ^■^	50.42 (42.83; 56.56)
**hsCRP (mg/L)**	4.05 (2; 5) ^†,^*	6.6 (3.11; 7.3)	10.85 (3; 12)	8.83 (5; 10.2)
**Cholesterol (mg/dL)**	135.3 (115; 150) ^¶,^*	165.8 (134; 187)	169.4 (134; 211) ^‡^	200.68 (154; 230)
**Triglycerides (mg/dL)**	108.43 (88; 102) ^#,^*	139.95 (102; 171.5)	182.21 (115.5; 215) ^●^	231.4 (160; 296)
**HbA1c (%)**	5.01 (4.9; 5.1) ^#,^*	7.21 (6.5; 7.65) ^♣^	8.38 (7.2; 9.6)	8.45 (7.8; 9)
**UACR (mg/g)**	14.67 (10.19; 17.25) ^#,^*	21.03 (15.10; 27.47) ^♣^	104.5 (60; 139.04) ^■^	1044.16 (464.89; 1365.47)
**uIL-17A/creat (pg/g)**	2.68 (1.94; 2.95) ^#,^*	7.64 (6.18; 9.01) ^♣^	13.89 (11.48; 16) ^■^	27.32 (22; 31.32)
**sIL-17A (pg/mL)**	4.89 (3.46; 5.54) ^#,^*	10.83 (9.17; 12.3) ^♣^	18.81 (17.48; 20.61) ^■^	37.8 (32.43; 42.3)
**uIL-18/creat (pg/g)**	31.94 (18.27; 48.97) ^∆,^*	45.53 (28.06; 60.44) ^♣^	89.44 (69.11; 109.1) ^■^	131.5 (94.18; 162.38)
**sIL-18 (pg/mL)**	83.5 (64.46; 102.17) ^#,^*	119.78 (94.17; 145.1) ^♣^	153.27 (120.42; 186.46) ^■^	222.03 (154.37; 261.93)
**uIL-10/creat (pg/g)**	15.1 (12.89; 16.38) ^#,^*	8.97 (6.98; 11.23) ^♣^	5.85 (4.72; 6.27) ^⁑^	4.86 (3.71; 5.81)
**sIL-10 (pg/mL)**	19.75 (17.45; 20.94) ^#,^*	14.07 (12.2; 16.11) ^♣^	11.45 (10.64; 11.98) ^■^	9.54 (8.06; 11.28)
**smtDNA**	15.71 (12.87; 17.78) ^#,^*	10.73 (8.21; 12.38) ^♣^	6.92 (5.35; 7.2) ^■^	3.63 (1.82; 4.52)
**umtDNA**	3.12 (1.08; 4.56) ^ↂ,^*	5.16 (2.79; 7.07) ^♣^	8.27 (6.57; 10.26) ^■^	12.59 (10.69; 14.1)
**Synaptopodin/creat (mg/g)**	10.1 (7.44; 11.21) ^#,^*	18.18 (15.38; 21.46) ^♣^	26.55 (24.74; 28.11) ^■^	79.33 (34.75; 133.56)
**Podocalyxin/creat (mg/g)**	38.7 (30.98; 49.55) ^#,^*	65.3 (58.43; 70.72) ^♣^	128.62 (114.42; 152.58) ^■^	520.98 (393.3; 620.54)
**KIM-1/creat (pg/g)**	39.3 (27.7; 46.7) ^#,^*	78.37 (66.8; 93.6) ^♣^	134.02(127.93; 147.68) ^■^	668.5 (595.32; 815.9)
**NAG/creat (ng/g)**	2.02 (1.65; 2.22) ^#,^*	4.74 (2.23; 5.96) ^♣^	12.66 (9.87; 16.09) ^■^	18.52 (16.38; 18.83)

Clinical and biological data are presented as medians and IQR for variables with skewed distribution. Significance between healthy controls and normoalbuminuric group, ^#^ *p* < 0.001; ^†^ *p* = 0.016; ^¶^ *p* = 0.001; ^∆^ *p* = 0.005; ^ↂ^ *p* = 0.004; significance between normoalbuminuric group and microalbuminuric group, ^⌂^ *p* = 0.008; ^♣^ *p* < 0.001; ^▲^ *p* = 0.012; significance between microalbuminuric group and macroalbuminuric group, ^♦^ *p* = 0.015; ^■^ *p* < 0.001; ^‡^ *p* = 0.012; ^●^ *p* = 0.008; ^⁑^ *p* = 0.001; significance between healthy controls vs. normoalbuminuric group vs. microalbuminuric group vs. macroalbuminuric group; * *p* < 0.001; BMI: body mass index; SBP: systolic blood pressure; DBP: diastolic blood pressure; DM: diabetes mellitus; eGFR: estimated glomerular filtration rate; hsCRP: high-sensitive C-reactive protein; UACR: urinary albumin/creatinine ratio; KIM-1/creat: urinary kidney injury molecule-1/creatinine ratio; NAG/creat: *N*-acetyl-β-(D)-glucosaminidase/creatinine ratio; Podocalyxin/creat: Podocalyxin/creatinine ratio; Synaptopodin/creat: Synaptopodin/creatinine ratio; Hb: haemoglobin; HbA1C: glycated haemoglobin; smtDNA: serum mitochondrial DNA (mtDNA/nDNA copies); umtDNA: urinary mitochondrial DNA (mtDNA/nDNA copies) normalized to urine creatinine; sIL: serum interleukin; uIL: urinary interleukin/creatinine ratio.

**Table 2 ijms-24-09803-t002:** Univariable regression analysis for serum and urinary mtDNA and UACR, eGFR, parameters of podocyte damage (synaptopodin and podocalyxin), and PT dysfunction (KIM-1 and NAG).

Parameter	Variable	R²	Coef β	*p*
**Urinary mtDNA**	**UACR**	0.255	0.0036	<0.001
	**eGFR**	0.474	−0.210	<0.001
	**Synaptopodin/creat**	0.307	0.064	<0.001
	**Podocalyxin/creat**	0.546	0.015	<0.001
	**KIM-1/creat**	0.499	0.011	<0.001
	**NAG/creat**	0.466	0.392	<0.001
**Serum mtDNA**	**UACR**	0.228	−0.004	<0.001
	**eGFR**	0.460	0.232	<0.001
	**Synaptopodin/creat**	0.252	−0.067	<0.001
	**Podocalyxin/creat**	0.422	−0.0156	<0.001
	**KIM-1/creat**	0.388	−0.011	<0.001
	**NAG/creat**	0.419	−0.4287	<0.001

mtDNA: mitochondrial DNA; UACR: urine albumin/creatinine ratio; eGFR: estimated glomerular filtration rate; KIM-1/creat: urinary kidney injury molecule-1/creatinine ratio; NAG/creat: *N*-acetyl-β-(D)-glucosaminidase/creatinine ratio.

**Table 3 ijms-24-09803-t003:** Univariable regression analysis for serum and urinary mtDNA and parameters of inflammation (ILs-17A, IL-18, and IL-10).

Parameter	Variable	R²	Coef β	*p*
**Urinary mtDNA**	**uIL-17A/creat**	0.490	0.301	<0.001
	**uIL-18/creat**	0.409	0.055	<0.001
	**uIL-10/creat**	0.375	−0.662	<0.001
**Serum mtDNA**	**sIL-17A**	0.552	−0.285	<0.001
	**sIL-18**	0.340	−0.041	<0.001
	**sIL-10**	0.530	0.931	<0.001

mtDNA: mitochondrial DNA; sIL: serum interleukin; uIL/creat: urinary interleukin/creatinine ratio.

**Table 4 ijms-24-09803-t004:** Multivariable regression analysis for serum and urinary mtDNA.

Parameter	Variable	Coef β	*p*	95% CI	Prob > F	R²
**Serum mtDNA**	**UACR**	−0.016	<0.001	−0.326 to −0.660	0.00001	0.626
	**KIM-1**	−0.215	0.068	−0.278 to −0.048		
	**sIL-17A**	−0.170	<0.001	−0.225 to −0.123		
	**sIL-10**	0.507	<0.001	0.337 to 0.677		
**Urinary mtDNA**	**uIL-18/creat**	0.013	0.021	0.002 to 0.024	0.00001	0.631
	**uIL-10/creat**	−0.182	0.073	−0.325 to −0.036		
	**Podocalyxin/creat**	0.008	0.027	0.005 to 0.117		
	**NAG/creat**	0.097	<0.001	0.010 to 0.183		
	**UACR**	0.029	0.023	0.016 to 0.178		
	**eGFR**	−0.430	<0.001	−0.367 to −0.529		

mtDNA: mitochondrial DNA; UACR: urine albumin/creatinine ratio; eGFR: estimated glomerular filtration rate; podocalyxin/creat: podocalyxin/creatinine ratio; KIM-1/creat: urinary kidney injury molecule-1/creatinine ratio; NAG/creat: *N*-acetyl-β-(D)-glucosaminidase/creatinine ratio; sIL: serum interleukin; uIL/creat: urinary interleukin/creatinine ratio.

## Data Availability

The data that support the findings of this study are available from the corresponding author upon reasonable request.

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
