# Peer review of "Mitochondrial DNA Changes in Blood and Urine Display a Specific Signature in Relation to Inflammation in Normoalbuminuric Diabetic Kidney Disease in Type 2 Diabetes Mellitus Patients"

_ijms, 2023, doi:10.3390/ijms24129803_

Round 1

Reviewer 1 Report

In this article, the authors fully explain that mtDNA changes impact podocytes, proximal tubules and associated inflammatory processes in normoalbuminuric DKD of type 2 DM patients. The experiments were well designed and performed whereas the manuscript is well prepared. However, there are still some recommendations.

Recommendations:

1.     Please list the full form when it is first mentioned. For example, full form should be stated for “DM” in the abstract.

2.     There are some errors in gramma and expression. For example, “This data shows significantly differences” in the third line of results; line 213 of the article “within the kidney or”; and the subject is absent in the first sentence of the fourth paragraph of introduction.

3.     Subtitle of Result 2.1 is missing.

4.     It would be easier to understand if the sentences from lines 213 to 218 of the article have been written more succinct.

5.     Some references are missing in the article. For example, line 199 of the article “Several studies showed that patients with type 2 DM and DKD had lower levels of serum mtDNA than diabetic control.”; line 222 “In the study by Wei et al., the authors remarked that higher levels of urinary supernatant mtDNA correlated with lower baseline eGFR”; line 257 “In an experimental study in mice, IL-17A blockade elicited a significant decrease in albuminuria and an improvement in renal lesions

6.     The last sentence of 3.1 of results mentioned that “However, due to lack of renal tissue analysis, impairment to intra-renal mtDNA cannot be excluded”. Please check reference [28] to determine whether renal tissue analysis has been done previously.

There are some errors in gramma and expression. 

Author Response

Point-by-point reply

Reviewer 1

Comments and Suggestions for Authors

In this article, the authors fully explain that mtDNA changes impact podocytes, proximal tubules and associated inflammatory processes in normoalbuminuric DKD of type 2 DM patients. The experiments were well designed and performed whereas the manuscript is well prepared. However, there are still some recommendations.

Recommendations:

  1. Please list the full form when it is first mentioned. For example, full form should be stated for “DM” in the abstract.

Thank you for your recommendation.

The full form of diabetes mellitus followed by “DM” has been introduced in the abstract.

  1. There are some errors in gramma and expression. For example, “This data shows significantly differences” in the third line of results; line 213 of the article “within the kidney or”; and the subject is absent in the first sentence of the fourth paragraph of introduction.

Thank you for your pertinent suggestion.

The corrections have been made, such as follows:

- Results 2.1 These data show significantly differences between the groups studied.

- Line 213 The authors queried the origin of urinary mtDNA, namely whether increased mtDNA in urine is related to a continuous destructive process of mitochondrial content within the kidney or this is due to rather than to an increased clearance of circulating mtDNA [28].

- Subject in the first sentence of the fourth paragraph: Among the Multiple mechanisms are involved in the development and progression of DKD.

  1. Subtitle of Result 2.1is missing.

      Thank you for your observation.

      The subtitle of Results 2.1 has been introduced.

2.1. Demographic, clinical, and biological data of patients with type 2 DM and healthy control subjects

  1. It would be easier to understand if the sentences from lines 213 to 218 of the article have been written more succinct.

Thank you for your recommendation.

The authors queried the origin of urinary mtDNA, namely whether increased mtDNA in urine is related to a continuous destructive process of mitochondrial content within the kidney or this is due to rather than to an increased clearance of circulating mtDNA [28]. Studies focused on alterations of serum mtDNA or urinary mtDNA did not include correlations with renal histology, except for the study by Wei et al. [28]. In this study, the authors showed that urinary supernatant mtDNA had a significant invers correlation with intra-renal mtDNA and correlated with severity of tubulointerstitial fibrosis, but not with glomerulosclerosis on biopsy, even in early stages of diabetic nephropathy [28].

  1. Some references are missing in the article.

      Thank you for your observation. The appropriate references have been introduced in the text.

      - line 199 of the article “Several studies showed that patients with type 2 DM and DKD had lower levels of serum mtDNA than diabetic control.” Ref 26;

      - line 222 “In the study by Wei et al., the authors remarked that higher levels of urinary supernatant mtDNA correlated with lower baseline eGFR” Ref 28;

      - line 257 “In an experimental study in mice, IL-17A blockade elicited a significant decrease in albuminuria and an improvement in renal lesions” Ref 29.

  1. The last sentence of 3.1 of results mentioned that “However, due to lack of renal tissue analysis, impairment to intra-renal mtDNA cannot be excluded”. Please check reference [28]to determine whether renal tissue analysis has been done previously.

Thank you for your valuable comment.

In our study we did not perform renal tissue analysis.

However, in our study, due to lack of renal tissue analysis, impairment to intra-renal mtDNA cannot be excluded.

In Ref 28 (Wei, et al), the authors have performed renal tissue analysis in order to study impairment to intra-renal mtDNA (line 218).

Comments on the Quality of English Language

There are some errors in gramma and expression. 

The errors in gramma and expression have been corrected according to the recommendations.

Reviewer 2 Report

In this manuscript authors evaluate MtDNA contents, markers of podocyte injury and PT dysfunction in early DKD of 136 type 2 DM patients.

1)           Sampling: in Methods, it is stated: “A total of consecutive 220 patients with type 2 DM attending the Outpatient Department of Nephrology and the Outpatient Department of Diabetes and Metabolic Diseases (from January 2021 through December 2022), aged between 50 and 78 years, were screened for the study according to person visits and chart reviews…”. As the selection criteria seem not to be so restrictive, it would be expectable to have more consecutive patients being recruited from a center of this natures in such a long period. What is the reason for this limited number of individuals? I am not discussing about power but about potential selection biases. Please explain the selection process in more detail. Were these patients consecutively included? Any other of sampling method applied?

2)           Blood pressure is commonly associated with albuminuria (micro and macro). It looks like values of SBP increase among the different categories. A mean of 148 mmHg in the severe albuminuric patients seems to be relevantly higher than 138 mmHg found as mean value in normo / mildly increased individuals. The impact of Hypertension on podocyte health may be relevant and should be discussed and clarified.

3)           In table 4, we must assume the provided p and R2 values are the “total model” ones. Now: a R2 of 0.62 or 0.63 for models like these are unusual, as correspond with R values of approximately 0.80. If these values are correct, authors should remark in the text and explain.

4)           Authors conclude: “Mitochondrial DNA changes in serum and urine display a specific signature in relation to inflammation both at podocyte and tubular level in normoalbuminuric type 2 DM patients, independently of renal function decline”. So: Do we have to assume mtDNA is not a good marker of renal decline? What may be its relevance in DKD? Any? None?

5)           Being a cross sectional study, it is very difficult to sustain the last conclusion: “independently of renal function decline”. Authors did not follow renal function along any period. The kinetics of mtDNA might differ among the different stages of evolution: we cannot strongly conclude about the prognostic relevance of a marker by considering  just one cross-sectional evaluation.

6) What is the “specific signature” the authors mention in the title? Can the authors explain what is specific for normoalbuminuric when compared with other categories they included into the study?

Author Response

Point-by-point reply

Reviewer 2

In this manuscript authors evaluate mtDNA contents, markers of podocyte injury and PT dysfunction in early DKD of 136 type 2 DM patients.

1)           Sampling: in Methods, it is stated: “A total of consecutive 220 patients with type 2 DM attending the Outpatient Department of Nephrology and the Outpatient Department of Diabetes and Metabolic Diseases (from January 2021 through December 2022), aged between 50 and 78 years, were screened for the study according to person visits and chart reviews…”. As the selection criteria seem not to be so restrictive, it would be expectable to have more consecutive patients being recruited from a center of this natures in such a long period. What is the reason for this limited number of individuals? I am not discussing about power but about potential selection biases. Please explain the selection process in more detail. Were these patients consecutively included? Any other of sampling method applied?

Thank you for your pertinent observation.

The patients were consecutively included in the study. The limited number of patients included in the study derives from the fact that the screening was very difficult to perform during the Covid-19 pandemic. Patients postponed their scheduled visits due to fear of contacting Covid-19. Also, patients who attended the visits were reluctant to provide written informed consent in order to be included in the study. These facts precluded the inclusion of a larger cohort of patients.

2)           Blood pressure is commonly associated with albuminuria (micro and macro). It looks like values of SBP increase among the different categories. A mean of 148 mmHg in the severe albuminuric patients seems to be relevantly higher than 138 mmHg found as mean value in normo / mildly increased individuals. The impact of Hypertension on podocyte health may be relevant and should be discussed and clarified.

Thank you for your valuable comment.

In our study, there were no statistically significant differences between the groups of patients studied with regard to hypertension.

There is well known that hypertension and the RAAS could interfere with podocyte function. However, all patients included in the study were on ACEIs or ARBs. Thus, these medications could not introduce a bias in the interpretation of mtDNA changes in relation to biomarkers of podocyte damage.

3)           In table 4, we must assume the provided p and R2 values are the “total model” ones. Now: a R2 of 0.62 or 0.63 for models like these are unusual, as correspond with R values of approximately 0.80. If these values are correct, authors should remark in the text and explain.

Thank you for your comment.

The R2 values are correct and correspond to R values of 0.8, a fact which shows that there are strong correlations between mtDNA and the variables studied.

4)           Authors conclude: “Mitochondrial DNA changes in serum and urine display a specific signature in relation to inflammation both at podocyte and tubular level in normoalbuminuric type 2 DM patients, independently of renal function decline”. So: Do we have to assume mtDNA is not a good marker of renal decline? What may be its relevance in DKD? Any? None?

Thank you for your valuable comment.

Altered mitochondrial deoxyribonucleic acid (mtDNA) under excessive oxidative stress conditions is of major importance in the pathogenesis of DKD. The relevance of mtDNA changes in serum and urine resides in the detection of these modifications early in the course of DKD, before the occurrence of albuminuria.

-The statement “independently of renal function decline” has been deleted in the abstract and conclusions.

- “These results were independent of renal function decline. -deleted in Discussion

5)           Being a cross sectional study, it is very difficult to sustain the last conclusion: “independently of renal function decline”. Authors did not follow renal function along any period. The kinetics of mtDNA might differ among the different stages of evolution: we cannot strongly conclude about the prognostic relevance of a marker by considering just one cross-sectional evaluation.

Thank you for your pertinent observation.

The variations of mtDNA in blood and urine were related to the albuminuria stage, relevant even in the normoalbuminuric type 2 DM patients.

In the paragraph which enumerates the limitations of the study there is specified that one of the limitations resides in its cross-sectional design. Therefore, the statement “independently of renal function decline” has been deleted in the abstract, discussion, and conclusions.

6) What is the “specific signature” the authors mention in the title? Can the authors explain what is specific for normoalbuminuric when compared with other categories they included into the study?

Thank you for your valuable comment.

In our study, the “specific signature” of mitochondrial dysfunction in normoalbuminuric type 2 DM patients relies on the fact that the deregulated pattern of mtDNA in serum and urine parallels inflammation both at glomerular and tubular level early in the course of DKD.